

# Design and evaluation of a simulated wound management course for postgraduate year one surgery residents

Xin Qi[1,*], Rui He[1,*], Bing Wen[1], Qiang Li[1] and Hongbin Wu[2]

[1] Plastic Surgery and Burn, Peking University First Hospital, Beijing, China
[2] Institute of Medical Education/National Centre for Health Professions Education Development, Peking University, Beijing, China
[*] These authors contributed equally to this work.

## ABSTRACT

**Background**. It is vital to cover wound management knowledge and operations in the early stages of resident training. With this in mind, a simulated wound management course for postgraduate year one surgery residents (PGY1s) was designed and its effectiveness was evaluated.

**Methods**. A retrospective quasi-experimental method was used. PGY1s in 2014 constituted the control group, and PGY1s in 2015 and 2016 constituted the intervention group. The course given to the control group comprised didactic teaching followed by deliberate practice plus immediate personalized feedback. The newly designed course given to the intervention group was reconstructed and disassembled into four components according to the simulation-based mastery learning model, which were baseline test, interactive learning, basic skills practice, and reflective learning. The same performance assessments were used in the control and intervention group, including process measurement and outcome measurement.

**Results**. The process measurement showed that the intervention group's scores were significantly higher in the "dissociation of subcutaneous tissue" and "quality of suturing and knots". The outcome measurement showed that the accuracy of debridement was greatly improved and both key and total suture numbers were significantly higher in the intervention group.

**Conclusions**. Simulation-based mastery learning was incorporated into our proposed course framework, promoting the learning outcome of PGY1s. It has the potential to be adapted for other surgical training sites for residents in China.

## INTRODUCTION

Traditional surgical resident training has long followed the classical Halstedian teaching model of "see one, do one, teach one". However, significant changes have occurred in the training of surgical residents in China with the broadening of simulation-based training (*Willis & Van Sickle, 2015*) and competency-based medical education (*Li & Wang, 2013*). In the non-threatening, confidential, and "psychologically safe" (*Fanning & Gaba, 2007*;

Corresponding authors
Xin Qi, 04983@pkufh.cn
Hongbin Wu, wuhongbin@pku.edu.cn

*Rudolph et al., 2008*) training environment of simulation labs, trainees can continually improve their surgical skills with deliberate practice (*Ericsson, 2008*; *McGaghie et al., 2011*).

Wounds vary from minor vulnera (e.g., abrasions, lacerations) to severe life- or limb-threatening injuries. Wound management is a challenge faced by surgeons. If not carefully dealt with, clinical interventions could result in adverse effects on the skin, deep soft tissues, and even organs. Surgical site infections are the most typical wound, with an incidence of 0.4%–17.8% (*World Health Organization, 2018*). Currently, wound management training is distributed across different disciplines with different patterns. Trainees are mostly specialists or nurses, while medical students and residents, as major frontline staff, have insufficient opportunities to participate in training (*Yim et al., 2014*; *Lupon et al., 2019*). Wound management is an essential and required basic skill for postgraduate year 1 residents (PGY1s) in surgery training, and it is necessary to create a simulation-based wound management course for PGY1s to avoid medical errors.

To effectively implement standardized training for surgical residents, Peking University First Hospital (PKUFH) established a surgery school in 2014, which was an important innovation in China (*Qi et al., 2016*). Plastic surgeons were assigned to instruct the simulation-based wound management course. The course content included basic skills and the concept of debridement and closure, which was delivered in the classic pattern of didactic teaching followed by deliberate practice. However, the 2014 summative assessment did not yield satisfactory results. We found that despite PGY1s had learned various basic technical skills during undergraduate training in China, they were unable to effectively integrate these skills when facing a complex simulated situation. Therefore, simple technical skills training was not suitable for the residents, and specific training requirements were identified to allow the trainees to train in a more clinically authentic environment, that is, decision-oriented procedural performance regarding wound management.

In this study, we designed and dynamically revised a wound management course by adapting simulation-based mastery learning (SBML) model aiming to improve the wound management ability of PGY1s. The objective of this article is to describe the development, implementation, evaluation and improvement of the course and investigate the effect of the course by assessing PGY1s' learning outcomes.

## MATERIAL AND METHODS

### Study design and participants

This was a retrospective quasi-experimental study. Participants of the wound management course were PGY1s at PKUFH in 2014, 2015, and 2016, excluding those who did not complete the course or the summative assessment. All PGY1s can enter the surgical training curriculum only if they have passed the surgical residency entry examination, the criteria of which remained the same for each year of the study. By the end of the course, learners were to be able to conduct debridement and ultimately close the wound. Course contents included basic skills (asepsis and instrument identification, knot tying, suturing, excision, debridement, dissociation, wound closure, skin flaps, etc.), and the knowledge of debridement and closure. The course contents had been collectively evaluated

and reviewed by experts in the surgery school to ensure their quality. In the skills lab, a simulated procedural training was adopted using the pattern of small group animal surgery. Instructors, examiners, learning objectives, and assessment criteria were consistent across the three years. The simulation-based exercises were carried out in the same skills lab, and the duration of the course was 240 min. The performance of two groups who completed both assessments were used to evaluate the impacts of different simulation-based courses. The overview of this study is described in Fig. 1.

The two groups are as follows:

1. Control group: In 2014, the course for PGY1s was organized as a 60-min didactic teaching session including an operative demonstration, followed by 180 min of deliberate practice plus immediate personalized feedback. A summary was given at the end. Rabbits under general anesthesia were used for procedural training.

2. Intervention group: Based on the results in 2014, the course was reconstructed to include four components of SBML: baseline test, interactive learning, basic skills practice, and reflective learning. The process emphasized learner-directed small group learning with instructor facilitation. A low-fidelity simulator was adopted and cadaveric pork belly skins were used for procedural training.

The study was approved by the Institutional Review Board of PKUFH (2018-123). The skills lab has the license issued by Beijing Municipal Commission of Science and Technology for the use of experimental animals.

## Simulation-based mastery learning intervention

According to the learning theory foundations of SBML (*McGaghie & Harris, 2018*), the course (for more details, see Supplement 1) was designed following three learning theory foundations: behavioral, constructivist, and social cognitive (Table 1).

## Component 1. Baseline Test (10 Minutes)

PGY1s were required to perform a complete excision of the "dumbbell" necrotic tissue within 10 min and close the wound (Fig. 2). Later, they were required to consider how to repair a defect with the shape of two connected triangles. Starting from the classic case of plastic surgery, clinical problems could stimulate their learning requirements and interests. Through the baseline test, instructors could have a deeper understanding of their knowledge, skills, and attitude so as to calibrate the learning direction and teaching strategies in the following parts.

## Component 2. Interactive Learning (60 Minutes)

PGY1s were encouraged to review their performance on the blackboard and share their thinking process and selection basis. Then instructors speculated about the PGY1s' mental approaches through observation and prepared their plans accordingly. The concepts of debridement and closure were communicated through narration and interrogation between instructors and PGY1s. The role of the instructor in this component was a facilitator.

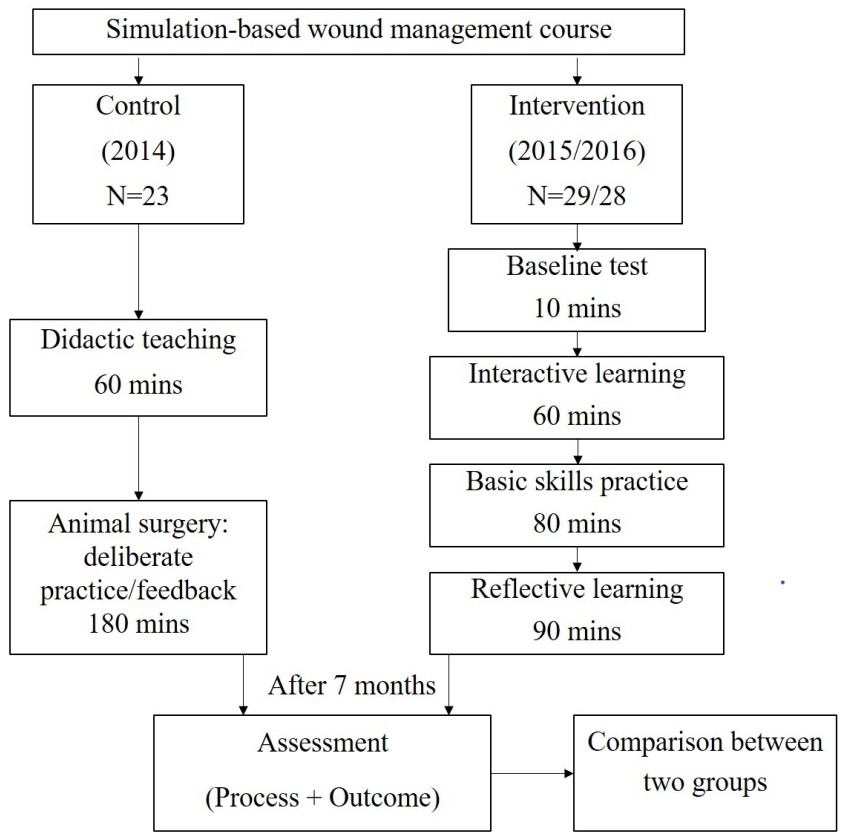

**Figure 1** The study overview.

**Table 1** Learning theory foundations of the course.

| Theory foundations | Wound management skills acquired and assessed using SBML |
| --- | --- |
| Behavioral | Asepsis and instrument identification, knot tying, suturing, excision, debridement, dissociation, wound closure, skin flaps, etc. |
| Constructivist | Recognize clinical signs, critical thinking, problem solving: identify a lesion's range and depth, develop a treatment plan, be aware of protecting skin and soft tissue, reduce wound tension, use local flap technique to repair the wound, etc. |
| Social Cognitive | Reflective learning, peer education (resident as instructor), increasing clinical self-efficacy about complicated wounds, acquiring communication and collaboration skills. |

## Component 3. Basic Skills Practice (80 Minutes)

PGY1s were expected to complete the closure of round, square, and triangular wounds. By adapting Peyton's four-step (demonstration, deconstruction, comprehension, and performance) approach (*Nikendei et al., 2014*), the instructor offered hands-on deliberate practice focusing on practical issues. In this component, the instructor mainly corrected

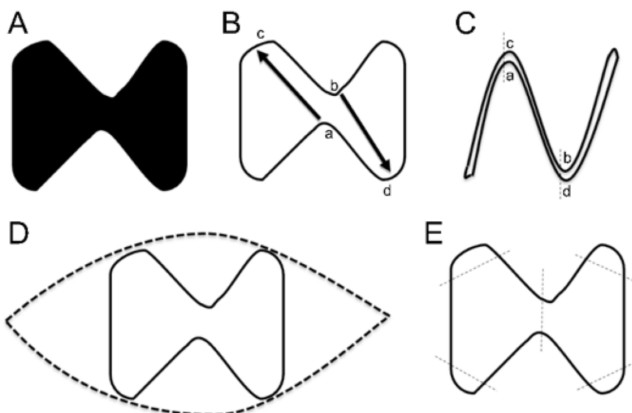

**Figure 2** **Baseline test.** (A) The necrotic area was marked. (B) The PGY1s were required to perform a complete excision of the lesion and a primary closure of the wound. They were expected to leave flaps a and b open and suture the two key points—point c to point a and point b to point d. (C) The expected results. (D) Typical error: spindle excision (normal tissue was not retained and incision tension was increased). (E) Typical error: no preoperative planning (cut mechanically along the edge, and only sutured the low tension areas).

their basic skills (behaviors), discerned possible problems, and assessed their understanding of Component 2.

## Component 4. Reflective Learning (90 Minutes)

The instructor provided feedback based on the completion of Component 3, expanded the wound closure content, focusing on the principles and concepts of various local flaps, as well as certain aesthetic issues (e.g., dog ear). Subsequently, PGY1s continued to complete round, square, and triangular wound closures, or more complex and challenging wounds. They were encouraged to communicate and collaborate with each other and offer help to others along with the instructor. Deficiencies in cognition and procedure were exposed based on their performances. The instructor observed the entire process and facilitated continuous reflection by the PGY1s.

## Performance assessment

Our course is part of the entire surgical training curriculum, so the summative assessment comprises one round of objective structured clinical examination (Fig. 3), which is held seven months after the course. PGY1s' performances were assessed from the aspects of both the process and the outcome. The examiners were three attending doctors from different departments, who received training before the assessment. In detail, an instrument with a 10-point rating scale modified from Objective Structured Assessment of Technical Skills (*Hopmans et al., 2014*) was used to assess the PGY1s' surgical operation process according to six aspects (maintaining a sterile field, knowledge and handling of instruments, quality of excision, quality of debridement, dissociation of subcutaneous tissue, and quality of suturing and knots). Three examiners graded the process of each resident, and average scores were obtained to serve as overall scores for the process measurement. The outcome measurement was made using a checklist focusing on five aspects (residual marking of

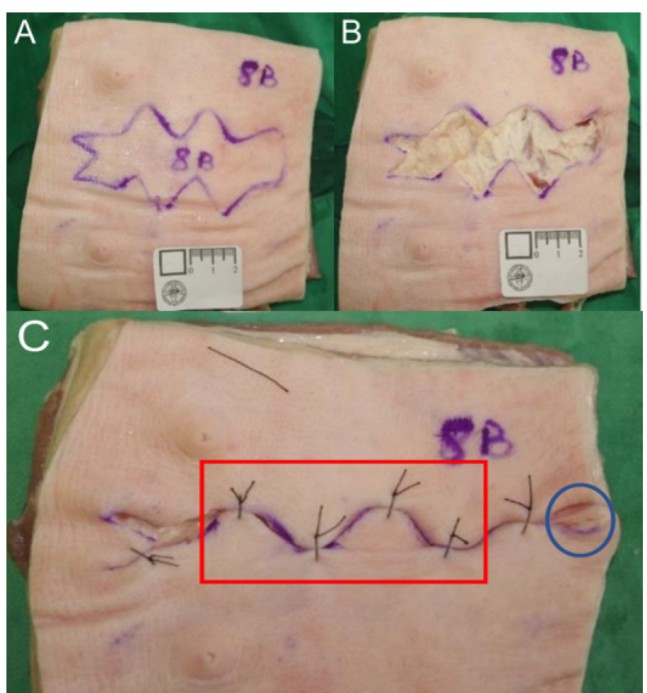

**Figure 3** **Outcome measurement.** (A) An irregular area of $10 \times 3.8$ cm², marked on cadaveric pork belly skin , was regarded as the necrotic part, affecting the deep fascia. The PGY1s were required to perform a primary suture after debridement within 15 min. (B) The PGY1 excised along the margin, but residual marking was visible, and wound bed wasn't deep enough to the fascia (C) The box showed four key sutures, and the cycle marked the dog ear.

incision margin, depth to deep fascia, spindle resection, dog ear treatment, subcutaneous suture), along with the recording of total suture number counts and key suture number counts. The same three examiners who graded the instrument of the process measurement also observed and discussed until consensus the marking of all the areas of the outcome measurement instrument (five aspects and suture number counts).

## Statistical analysis

All results are expressed as mean and standard deviation (mean $\pm$ SD). Categorical variables were analyzed with a Chi-square test, nonnormally distributed continuous variables were analyzed with a Mann–Whitney test, normally distributed continuous variables were analyzed with a one-way ANOVA, and pairwise comparison was performed using a least significant difference test. The statistical analysis was performed using SPSS 22.0 software (IBM Corporation, USA). A $p$-value of less than 0.05 was regarded as statistically significant.

## RESULTS

### Participants

In 2014, 2015, and 2016, 31, 39, and 37 PGY1s were included, respectively, but only 23, 29, and 28 completed the course (Table 2). There were no significant differences in their ages ($p = 0.242$, $F = 1.447$).

### Baseline test

The wound closure performance of the PGY1s trained in 2015 was better than those trained in 2016. Furthermore, spindle resection occurred significantly less frequently among the 2015 PGY1s than the 2016 PGY1s (Table 3).

### Performance assessment

#### Process measurement

The results indicate the global rating scale scores of PGY1s presented an increased trend in scores over the three study years (for more details, see supplement 2). The scores of the 2015 and 2016 PGY1s were significantly higher than those of the 2014 PGY1s for the quality of debridement and dissociation of subcutaneous tissue. None of the 2014 PGY1s performed the dissociation of subcutaneous tissue, so they scored zero for this dimension. Based on the assessment results of 2015, dissociation of subcutaneous tissue training was reinforced for the 2016 PGY1s, and they got a higher score than 2015 PGY1s (Fig. 4).

#### Outcome measurement

Compared with those in 2014, the PGY1s in 2015 and 2016 showed no obvious improvement in the residual marking of incision margins (Table 4). However, their debridement accuracy (no spindle resection and more depth of deep fascia) did improve greatly. Both key and total suture numbers were significantly higher. Nobody in 2014 or 2015 dealt with dog ear treatment, but some PGY1s in 2016 did. However, under the premise of an increased choice of excision along the remarking and dissociation of subcutaneous tissue, the PGY1s in 2015 and 2016 aimed to close the wound as soon as possible and did not choose a subcutaneous suture.

## DISCUSSION

Mastery learning is a rigorous approach to competency-based education that requires students to engage in educational activities that have clear learning objectives and through deliberate practice to reach the minimal passing standard before advancing to another unit. SBML is in perfect alignment with such an educational philosophy; it significantly improves clinical skills for all participants and leads to skills retention (Motola et al., 2013). Although simulation-based education was adopted before 2014 in the PKUFH, the concept of mastery learning was introduced in 2015 and 2016, and the effect has significantly improved. In the meantime, the learning process has become self-paced, proactive, and self-reflecting (Bandura, 1997).

Emergency departments take in many skin and soft tissue (SST) injuries (Health Protection Agency, 2005; Nawar, Niska & Xu, 2007; Jones et al., 2012). An estimated 82.8%

**Table 2 Demographic information of the postgraduate year 1 surgery residents trained from 2014 to 2016.**

| | Control group | Intervention group | | p-value |
|---|---|---|---|---|
| | 2014 | 2015 | 2016 | |
| Gender:no.(%)male | 23/23(100) | 29/29(100) | 26/28(92.9) | 0.200[*] |
| Age:mean $\pm$ SD(year) | 24.78 $\pm$ 1.70 | 24.55 $\pm$ 1.21 | 25.25 $\pm$ 1.78 | 0.272[**] |

**Notes.**
[*] p value was calculated by the Chi-square tests
[**] p value was calculated by one-way ANOVA

**Table 3 Baseline test results of the postgraduate year 1 surgery residents trained in 2015 and 2016.**

| Dimensions | 2015 ($n = 34$) | 2016 ($n = 30$) | p-value |
|---|---|---|---|
| C [% (n)] | 47.0 (16) | 10.0 (3) | 0.001 |
| D [% (n)] | 11.8 (4) | 43.3 (13) | 0.004 |
| E [% (n)] | 41.2 (14) | 46.7 (14) | 0.659 |
| Wound closure [% (n)] | 11.8 (4) | 0 (0) | 0.116 |

**Notes.**
C, local flap transfer; D, spindle resection; E, necrosis resection, aimless and convenient suture.
p values were calculated by the Chi-square tests.

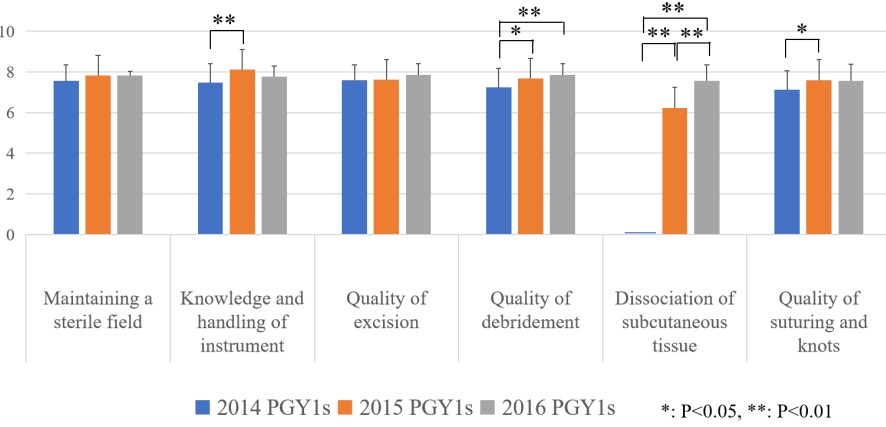

**Figure 4 Process measure.** p values were calculated by the least significant difference tests.

of traumatic injuries are accompanied by soft-tissue injuries, complicating potential limb salvage in patients (*Madubueze et al., 2011*). SST is both the starting and ending point of almost all surgeries. From a technical skill and equipment perspective, soft-tissue coverage procedures can be performed effectively even in low-resource settings (*Wu et al., 2016*). Therefore, as novices, it is necessary for PGY1s to receive early training for complex, rare, and critical SST wounds in skills labs in order to identify key issues and initiate proper preliminary treatment.

Today, surgical training places most weight on technical skills training, simulation, and learning by doing. The course in our study comprises mainly basic surgical skills except for local flap, but simple technical training cannot meet the clinical requirements of PGY1s.

**Table 4  Outcome measurement.**

| Dimensions | | 2014 | 2015 | 2016 | p-value | | |
|---|---|---|---|---|---|---|---|
| | | | | | 2014 vs. 2015 | 2014 vs. 2016 | 2015 vs. 2016 |
| Residual [% (n)] | | 30.4 (7) | 34.5 (10) | 28.6 (8) | 0.757[*] | 0.884[*] | 0.631[*] |
| Depth [% (n)] | | 30.4 (7) | 82.8 (24) | 71.4 (20) | 0.000[*] | 0.004[*] | 0.308[*] |
| Spindle [% (n)] | | 30.4 (7) | 0.0 (0) | 0.0 (0) | 0.002[*] | 0.002[*] | / |
| Dog ear [% (n)] | | 0.0 (0) | 0.0 (0) | 14.3 (4) | / | 0.117[*] | 0.052[*] |
| Subcutaneous [% (n)] | | 56.5 (13) | 10.3 (3) | 14.3 (4) | 0.000[*] | 0.001[*] | 0.706[*] |
| Numbers [n (min, max)] | Key numbers | 0(0, 4) | 3 (0, 4) | 3 (0, 4) | 0.000[**] | 0.014[**] | 0.702[**] |
| | Total numbers | 3 (0, 12) | 5 (0, 14) | 5 (0, 12) | 0.019[**] | 0.022[**] | 0.917[**] |

**Notes.**

Residual, residual marking of incision margin; Depth, depth to deep fascia; Spindle, spindle resection; Dog ear, dog ear treatment; Subcutaneous, subcutaneous suture; Numbers, Suture numbers.

[*]p values were calculated by the Chi-square tests

[**]p values were calculated by the Mann–Whitney tests

Since 2015, we have implanted questions into typical clinical cases in the baseline test to facilitate active study for PGY1s. After the baseline test, their perception, interpretation, and construction of meaning were motivated by authentic problems (*McGaghie & Harris, 2018*). Naturally, the learning model changed from didactic teaching to situated interactive learning, and the instructor's role changed to facilitator. Through interactive learning, PGY1s can benefit from the development of their cognitive processes and improve the efficiency and effectiveness of their behavioral skill acquisition by observing the practice of others (*Ericsson & Pool, 2016*). Decision making was introduced into interactive learning and gradually applied in basic skills practice and reflective learning for further training of clinical reasoning. The procedure was deconstructed into deliberate practice with basic surgical skills and experiential learning of repair and reconstruction concepts, so that the degree of difficulty gradually increased. Specific, informative feedback increased PGY1s' skills performance in a controlled setting (*Issenberg et al., 2005*). Through grasping and transforming experiences, PGY1s' competence improved. Therefore, this new course has the basic elements of SBML (*McGaghie et al., 2010*), which are the baseline test (with units sequentially ordered by increasing difficulty), engagement in educational activities, measurement of whether outcomes meet or exceed the mastery standard, and deliberate practice.

In SBML, deconstruction means to decompose the sophisticated skill into simple tasks to lower the cognitive load. Interactive learning focuses on the concept of debridement and closure through experiential learning, intending to establish PGY1s' clinical reasoning about wound management. Basic skills practice focuses on surgical skills training. Reflective learning is for reconstruction, manifested as concrete experience-reflection-active experimentation. Through the process of deconstruction and reconstruction, the overall cognitive load will decrease with practice, as some components of the skill begin to become automatic, which will transform into competence (*Motola et al., 2013*).

Furthermore, although wound tension reduction was included in the course content, it was submerged in an overwhelming amount of information in 2014, causing no PGY1s

to attempt dissociation of subcutaneous tissue. Since 2015, however, deconstruction of the original course content helped strengthen this aspect. The necessity of subcutaneous suture decreased as a direct consequence of the reduced wound tension, which benefited from excision along the remarking and dissociation of subcutaneous tissue, and therefore, most 2015 and 2016 PGY1s omitted this step as it was no longer necessary. PGY1s in 2015 and 2016 achieved significant improvement in dissociation of subcutaneous tissue.

Instructors can adjust the content, difficulty, and complexity of the simulation-based intervention in real time according to the baseline test. Although the hospital's resident selection criteria remain constant every year, residents may perform differently even in basic skills depending on their learning experience and educational background (*Barsuk et al., 2012*). This phenomenon was revealed in the baseline test results, which showed that the 2016 PGY1s' basic understanding of wound management was relatively inadequate compared with that of the 2015 PGY1s. In light of this information, the instructors were required to provide opportunities for individualized learning by dynamically adjusting the course content. An individual's level of prior knowledge may affect their learning and teaching methods, so PGY1s with high levels may afford and require more self-regulation and reflection, whereas PGY1s with low levels would benefit from more guidance (*Chernikova et al., 2020*). In 2015, more time was allotted for training in repair and reconstruction skills, whereas in 2016 the predetermined course content was completed, leaving more time to review basic concepts. We also revised the relevant course content based on the results of the previous year to check for deficiencies and fill in the gaps in each iteration. Debridement and subcutaneous dissociation were emphasized in 2016, and process measurement was changed accordingly. Meanwhile, with the continuous improvement of the course content, both key and total suture numbers were significantly higher in 2015 and 2016, indicating that both the surgical proficiency and the holistic view of wound closure had been obviously improved; some PGY1s in 2016 even managed the dog ear. Therefore, although there were some differences in the baseline test between PGY1s in 2015 and 2016, their performance assessment was approximately equal.

This study adopted a low-technology task training simulation. The effectiveness of the simulation depends on the demands of the clinical task. It is important that simulations can capture or represent a variety of patient problems and conditions (*Issenberg et al., 2005*). Simulations for novice trainees may not require simulators with high mechanical fidelity or simulations that are overly complex (*Motola et al., 2013*). For PGY1s to master the basics of flap design and implementation, porcine skin maybe the most cost-effective and efficient choice (*Hassan, Hogg & Graham, 2014*). The materials are relatively cheap and easy to obtain. Such simulations (using pork belly skin, for example), which are low in authenticity, can achieve high quality outcomes, possessing the potential to be transferred to courses in other surgical training programs in China.

## Limitations

First, the study was conducted in a surgical training site for residents with a small number of participants. Second, as the baseline test was not included in 2014, the actual baseline of the control group before the intervention was unknown. We were more concerned about

the consistency of outcome standards than entry standards. Third, the Mastery Education in Medicine (*Cohen et al., 2015*) model was not referred to at the beginning of the process of curriculum design. As our new course is still in the exploratory stage, future studies will aim to be designed strictly according to mastery learning standards by setting minimum passing scores and allowing continued repeated practice or study of an educational unit until reaching mastery, and post-intervention impacts of the course should be further improved and evaluated with the outcomes measured in the Kirkpatrick model levels 3 and 4. Fourth, the summative assessment was conducted 7 months after the intervention; although PGY1s had few opportunities to manage clinical cases of complex wounds during this period, there still could be certain effects on the ability of wound management for PGY1s. Finally, PGY1s were assessed in a skills lab, not in an actual clinical situation. Future studies may attempt to develop this as a trustable professional activity and conduct the assessments in the authentic clinical settings.

## CONCLUSIONS

Wound management is relatively stable and requires further improvement Our proposed course framework, which incorporated SBML, could promote the wound management competence of PGY1s. Given its effectiveness and feasibility, it has the potential to be adapted in other surgical training sites for residents in China.

**Definitions**

| | |
|---|---|
| **Dumbbell** | Flat shape with big ends and small middle, similar to dumbbell. |
| **Local flap** | Consists of skin and subcutaneous tissue that is harvested from a site nearby a given defect while maintaining its intrinsic blood supply. |
| **Deep fascia** | A layer of dense connective tissue that can surround individual muscles and groups of muscles to separate into fascial compartments. |
| **Dog ear** | A one-sided mound of redundant tissue, which is seen after the repair of certain skin lesions and defects. |
| **Debridement** | Removal of dead, damaged, or infected tissue to improve the healing potential of the remaining healthy tissue. |
| **Spindle resection** | Surgical removal of the target tissue using an incision with the shape of a slender round rod with narrowed ends. |

## ACKNOWLEDGEMENTS

The authors acknowledge the PGY1s and staffs at PKUFH for their cooperation and support.

### Funding

This work was supported by the Project of Medical Education Research from the Medical Education Branch of the Chinese Medical Association & Medical Education Professional Committee of China Association of Higher Education (2018B-N05035) and the National Natural Science Foundation of China for Young Scholars (71804005). The funders had no role in study design, data collection and analysis, decision to publish, or preparation of the manuscript.

### Grant Disclosures

The following grant information was disclosed by the authors:
Medical Education Branch of the Chinese Medical Association & Medical Education Professional Committee of China Association of Higher Education: 2018B-N05035.
National Natural Science Foundation of China: 71804005.

### Competing Interests

The authors declare there are no competing interests.

### Author Contributions

- Xin Qi conceived and designed the experiments, performed the experiments, analyzed the data, prepared figures and/or tables, authored or reviewed drafts of the paper, and approved the final draft.
- Rui He performed the experiments, prepared figures and/or tables, and approved the final draft.
- Bing Wen, Qiang Li and Hongbin Wu conceived and designed the experiments, authored or reviewed drafts of the paper, and approved the final draft.

### Human Ethics

The following information was supplied relating to ethical approvals (i.e., approving body and any reference numbers):

The Institutional Review Board of Peking University First Hospital provided approval to carry out the study within its facilities (2018-123).

### Data Availability

Raw measurements are available in the Supplemental Files.

### Supplemental Information

Supplemental information for this article can be found online at http://dx.doi.org/10.7717/peerj.11104#supplemental-information.

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
