# Peer review of "Design and evaluation of a simulated wound management course for postgraduate year one surgery residents"

_PeerJ, doi:10.7717/peerj.11104_

## Round 0.1 · original submission · Minor Revisions

Thank you for continued patience in reviewing process. As you know writing for publication is often an iterative process. Your manuscript has been reviewed by three very well accomplished scholars in the field of simulation based clinical education. Please pay careful attention to the reviewers suggestions. In particular it will be imperative to address the methodological and statistical analysis request and comments made by reviewer number one.

·

Basic reporting

Overall this article meets all standards. I do think that there may be room to define specific surgical jargon that is used throughout the article. Since this journal is not specifically targeting surgeons as a audience but rather has a broader audience there may need definitions provided as to what is meant by some of the terms used (i.e. dumbbell, dog ear treatment, spindle resection, etc...).
Also reference used to articulate the claim that clinical interventions could result in adverse effects on the skin, deep soft tissues ..with an average incidence of 2.6% needs to be updated. The citation used is from 1999.

Experimental design

Meets standards.
Define Peyton's four-step approach.

Validity of the findings

Please explain the use of Chi-square test and Mann-Whitney test.
would like p values and test use for baseline test provided in table 3
Missing information in figure 4 to be able to understand it. Please include what * means? Also p values needed.
Table 4 needs notes so that one can understand what *, triangles mean as subscripts. Also appears that the analyses were done at the group level, but this analysis seems to be more appropriately done with paired tests, specifically ANOVA. I am also not completely clear on what the percentages are ... are the percentage of participants who were successful? Based on the narration the outcome was suture numbers and +/- dog ear approach... but this does not seem to be reflected in the table outside of a %. Also how was decreased incision tension measured? This does not seem clearly measured in terms of a percent. Narration does not match well information provided in the table. Please revisit.

Additional comments

Thank you for your submission. This is an interesting study of importance to the field. There are however some issues with methods used for analysis as well as some of the jargon that needs addressing. It is possible that the methodological an be addressed for a successful publication, but give the fact that this was a retrospective study it is unclear if this is possible. I have left other comments in the sections above. Thank you for your submission and work on this important topic. I am not sure you really established that what you presented as study findings are evidence of mastery of suturing. It is possible that they are, but you need to more clearly set this up / speak to how they do.
I think the second paragraph of the discussion belongs in the background section. The description of the low tech simulation used included in the in the discussion belongs in the methods section.

Reviewer 2 ·

Basic reporting

The paper is well written and clear for readers. Those relevant literatures are cited properly to support the key message.

Experimental design

The study used the quasi-experimental design to compare the impact of the training course for wound management. It is not ideal but fits the real-world nature of program development for the course. The study outcome is well defined and relevant for clinical practices. But this study also didn't address the outcomes of Kirkpatrick model level 3 and 4 for clinical impact which is common weakness for many similar simulation studies.

Validity of the findings

The study results are presented well and showed the impact as expected. Due the limitation of study design, not sure the results will add more evidences of existing literate related to the topic. As many well-designed clinical trials have shown the post impact of simulation training in various specialty and programs. E.g., 2011 JAMA Systematic Review by Cook. https://pubmed.ncbi.nlm.nih.gov/21900138/

Do you know how many surgical procedures those learners did during those 7 months period before assessment? How do you know final performance variation is solely due to difference teaching methods and not due to exposure real clinical practice. This might need to be addressed in the discussion section.

Additional comments

Instead of using the parallel study flow chat, I would suggest to replace Figure 1 with flowchart to demonstrate longitudinal nature of the study design to reduce the confusion.

If you have any survey data from the learners about those courses will be very beneficial to support your conclusion.

The 1-3 paragraph of discussion section could be reduced with more concise language.

Reviewer 3 ·

Basic reporting

I find the basic reporting of the study to be very descriptive and realistic. It made
me feel that I had a good understanding of what kind of simulation training course was being conducted. I feel that most of the time, the authors has prepared this manuscript targeting to a more broad audience (rather than the Chinese medical education scholars).

The authors provided descriptive illustration towards the "traditional" way of surgical wound management course training for the residents, and how the new wound management course was being designed and implemented (although it was being mentioned mostly in the supplement material).

Nevertheless, I encourage the authors to consider the way the study is being reported regarding the following issues.

1. In Lines 128 - 133, it will be good to anchor this part with the designated objectives in your description, so that it can clearly represent the element of Mastery Learning. You could state that details see supplement.

2. Regarding Performance Assessment (Lines 143 - 159), are the residents allowed to practice and retake the exam (with the same MPS) if they did not pass the test?

3. How exactly is the MPS being determined? Could you provide a bit more information on that since it is an important element of ML.

4. Certain minor language errors identified, such as "carefully" in Line 57, a typo in Line 159 "toutcome".

Experimental design

This novel experiment is within the scope of the PeerJ Journal. Given the study is a quasi-experiment study, I understand that it would be difficult to control certain variables

I believe that this paper adds to the current literature in best practices in wound management towards surgical training, which few prior reports have been made public, especially those that have certain degree of alignment with the Mastery Learning (ML) Model in current medical education literature.

I am glad to see that the authors obtained IRB approval regarding the study. Just a quick question, do the authors have consents from the participating residents? Whether it's oral or written consent?

Validity of the findings

No comment.

---

## Round 0.2 · accepted · Accept

Thank you for your diligence as it relates to the revision of this manuscript. The reviewers are satisfied with your revisions. Also thank you for the very thorough rebuttal document.

Reviewer 2 ·

Basic reporting

No more comments.

Experimental design

No more comments.

Validity of the findings

No more comments.

Additional comments

Thank you for make modifications for those suggestions. The manuscript is great improved. Good work!

Reviewer 3 ·

Basic reporting

After proofreading as suggested, the manuscript is now more friendly to read towards international audience. Also it is good to see the authors have updated the references for the background session.

Experimental design

The authors have included the part regarding obtaining oral consent towards the residences.

Validity of the findings

no comment

Additional comments

Although the study, by its nature is not actually a mastery learning study, but the results of the study do support the idea towards shifting towards mastery learning do provide better learning outcomes. Given the significance of the study results, I believe that this manuscript is okay to be published, in hope to raise better attention internationally towards simulatio-based mastery learning model in procedural-based training.